# Fractal Analysis on Pore Structure and Modeling of Hydration of Magnesium Phosphate Cement Paste

Yuxiang Peng [1], Shengwen Tang [1,2,3,*], Jiasheng Huang [4], Can Tang [5], Lei Wang [6] and Yufei Liu [7]

1   State Key Laboratory of Water Resources and Hydropower Engineering Science, Wuhan University, Wuhan 430072, China; yxpeng@whu.edu.cn
2   State Key Laboratory of Building Safety and Built Environment, Beijing 100013, China
3   National Engineering Research Center of Building Technology, Beijing 100053, China
4   Department of Architecture and Civil Engineering, City University of Hong Kong, Kowloon, Hong Kong 999077, China; jshuang9-c@my.cityu.edu.hk
5   Department of Hydraulic Engineering, Tsinghua University, Beijing 100084, China; tang-c21@mails.tsinghua.edu.cn
6   College of Materials Science and Engineering, Xi'an University of Architecture and Technology, Xi'an 710000, China; wanglei535250684@xauat.edu.cn
7   Department of Applied Mathematics, Xi'an Jiaotong-Liverpool University, Suzhou 215123, China; yufei.liu18@student.xjtlu.edu.cn
*   Correspondence: tangsw@whu.edu.cn

**Abstract:** Magnesium phosphate cement (MPC) paste is hardened by the acid–base reaction between magnesium oxide and phosphate. This work collects and evaluates the thermodynamic data at 25 °C and a pressure of 0.1 MPa and establishes the hydration reaction model of MPC pastes. The influence of the magnesium–phosphorus molar (M/P) ratio and water-to-binder (W/B) ratio on the hydration product is explored by the thermodynamic simulation. Following this, the initial and ultimate states of the hydration state of MPC pastes are visualized, and the porosity of different pastes as well as fractal analysis are presented. The result shows that a small M/P ratio is beneficial for the formation of main hydration products. The boric acid acts as a retarder, has a significant effect on lowering the pH of the paste, and slows down the formation of hydration products. After the porosity comparison, it can be concluded that the decreasing of M/P and W/B ratios helps reduce porosity. In addition, the fractal dimension $D_f$ of MPC pastes is positively proportional to the porosity, and small M/P ratios as well as small W/B ratios are beneficial for reducing the $D_f$ of MKPC pastes.

**Keywords:** magnesium phosphate cement; fractal dimension; thermodynamic modeling; hydration products; porosity

## 1. Introduction

Magnesium phosphate cement (MPC), also known as chemically bonded ceramics [1], is usually made by dead-burned magnesium oxide with phosphate. It is commonly used in the quick repair of buildings and road traffic [2], solidification of nuclear waste and toxic substances [3], wastewater treatment [4], and restoration of teeth and bones in biology and medicine [5]. Compared with ordinary Portland cement, MPC has a shorter setting time, higher early strength, and better volume stability [2,6–8]. In the application of MPC, potassium dihydrogen phosphate (KDP) and ammonium dihydrogen phosphate (ADP) are usually used as the phosphate components. Such MPC added with KDP and ADP are short for MKPC and MAPC, respectively. The main reaction products of MKPC and MAPC-based materials are $MgKPO_4 \cdot 6H_2O$ (also called K-struvite) and $MgNH_4PO_4 \cdot 6H_2O$ (struvite) [9]:

$$MgO + KH_2PO_4 + 5H_2O \rightarrow MgKPO_4 \cdot 6H_2O \tag{1}$$

$$MgO + NH_4HPO_4 + 5H_2O \rightarrow MgNH_4PO_4 \cdot 6H_2O \tag{2}$$

The hydration process of MPCs is essentially an exothermic reaction of acid–base neutralization. It is believed that the hydration process of MPC can be divided into several steps: (a) MgO dissolves in water and releases $Mg^{2+}$ and $OH^-$; (b) $Mg^{2+}$ reacts with water molecules to form a positively charged hydrosol; (c) the hydrosol subsequently forms hydrogen phosphate with hydrogen phosphate ions; (d) the newly generated hydrogen phosphate forms a molecular network in the aqueous solution and then forms a gel [10–12].

Since the performance of MPC-based materials is mainly determined by the composition of their hydration products, there is a large number of studies describing the role of the water-to-binder mass ratio (W/B ratio) and the magnesium–phosphorus molar ratio (M/P ratio) in the hydration process of materials [13–15]. Xu et al. confirmed the influence of M/P in MKPCs and also concluded that potential hydration products were $MgHPO_4 \cdot 3H_2O$, $Mg_2KH(PO_4)_2 \cdot 15H_2O$ and K-struvite and that there are also unreacted MgO particles in MKPC system. Lahalle et al. [16,17] pointed out that under the experimental conditions of diluting suspension, boric acid retarded the hydration of MKPC, and at the same time, boron did not precipitate in the cement hydrate, nor did it become adsorbed on the surface of MgO particles, but stayed in the solution. However, for recent studies on MAPC-based materials, Sugama et al. [18] pointed out that the main hydration products of MAPC pastes are struvite, $Mg_3(PO_4)_2 \cdot 4H_2O$, and a small amount of $Mg(OH)_2$ and $MgHPO_4 \cdot 3H_2O$. Abdelrazig et al. [19] showed that $(NH_4)_2Mg(HPO_4)_2 \cdot 4H_2O$ was an intermediate product and continued to react to form struvite.

Commonly speaking, cement-based materials such as MPC pastes are considered porous composite materials and have an extremely complex pore structure, which largely determines their mechanical properties [20–22]. Recently, fractal analysis has been widely applied in revealing the microstructure properties in different materials [23–25], and fractal dimension is proved to be a vital parameter from fractal analysis. Jin et al. [26] showed that the relationship between fractal dimension and capillary pore volume was a negative function. Zhang et al. [27] considered that the fractal dimension of concrete could be estimated by the box-counting method, which is able to be applied on MPC-based materials to provide a further evaluation of the complexity of the pore structures.

At present, the studies on the mechanism of boric acid retardation in MPC-based materials and the relationship between hydration products and pore structure need to be further deepened, and computer simulations on the hydration mechanism of MPC-based materials can potentially serve as a supplement to experimental results of the hydration process of the MPCs [28–32]. This work establishes the hydration reaction model of MPC pastes and presents the method of thermodynamic calculation to study the evolution of hydration products with boric acid, as well as the relationship between the hydration products and pore structure of MAPC and MKPC pastes. Furthermore, fractal analysis is applied on the pore structure of the MPC pastes to evaluate the relationship between the microstructure and M/P or W/B ratios.

## 2. Methods and Algorithms

### 2.1. Establishment of a Thermodynamic Database of MPC

Now commonly used thermodynamic simulation software include PHREEQC, GEMS [33], and GWB [34], etc. In this work, PHREEQC is used to stimulate the hydration products of MPC pastes. PHREEQC is designed to perform a wide variety of aqueous geochemical calculations. Before performing thermodynamic simulation, relatively complete and reliable thermodynamic databases must be established. A comparison of different thermodynamic databases is shown in Table 1. During the modeling through PHREEQC with PITZER aqueous model, MINTEQA2 V.4 database was used to deal with the conditions of high ionic strength and complicated ionic interaction in MPC pastes for its extensive ion reaction data in the database. As summarized in Table 2, most of the thermodynamic data for MPC pastes were collected from the literature; others were from MINTEQ databases, where K was the chemical equilibrium constant.

**Table 1.** A Comparison of different thermodynamic databases.

| Database | Content(s) | Software Support(s) | Reference |
|---|---|---|---|
| Thermoddem | Liquid ions, gas phases, and mineral phases | PHREEQC, Crunch, Geochemist's workbench, Toughreact, Chess | [35] |
| PSI/Nagra | Solutes, actinides, and radioactive fission products in natural water | GEMS, PHREEQC | [36] |
| Blanc cement chemistry database | Detailed C-S-H data of different forms at different temperature | PHREEQC | [37] |
| MINTEQA2 V.4 database | A large number of complex ions and precipitation reactions | MINTEQ, Geochemist'S, WorkBench, PHREEQC | [38] |

**Table 2.** Thermodynamic data of MPC pastes.

| Ions and Phases | Reaction | Log K |
|---|---|---|
| $MgO$ | $MgO + 2H^+ = Mg^{2+} + H_2O$ | 21.8 [37] |
| $KH_2PO_4$ | $KH_2PO_4 = K^+ + H_2PO_4^-$ | 0.278 |
| $NH_4H_2PO_4$ | $NH_4H_2PO_4 = NH_4^+ + 2H^+ + PO_4^{3-}$ | −18.5 [39] |
| $MgNH_4PO_4 \cdot 6H_2O$ | $MgNH_4PO_4 \cdot 6H_2O = Mg^{2+} + NH_4^+ + PO_4^{3-} + 6H_2O$ | −13.13 [40] |
| $(NH_4)2Mg(HPO_4)_2 \cdot 4H_2O$ | $(NH_4)_2Mg(HPO_4)_2 \cdot 4H_2O = Mg^{2+} + 2NH_4^+ + 2PO_4^{3-} + 2H^+ + 4H_2O$ | −36.3 |
| $MgKPO_4 \cdot 6H_2O$ | $MgKPO_4 \cdot 6H_2O = Mg^{2+} + K^+ + PO_4^{3-} + 6H_2O$ | −10.62 [41] |
| $MgHPO_4 \cdot 3H_2O$ | $MgHPO_4 \cdot 3H_2O + H^+ = Mg^{2+} + H_2PO_4^- + 3H_2O$ | 1.41 [41] |
| $Mg_2KH(PO_4)_2 \cdot 15H_2O$ | $Mg_2KH(PO_4)_2 \cdot 15H_2O = 2Mg^{2+} + K^+ + H^+ + 2PO_4^{3-} + 15H_2O$ | −29.54 [41] |
| $Mg_3(PO_4)_2 \cdot 22H_2O$ | $Mg_3(PO_4)_2 \cdot 22H_2O = 3Mg^{2+} + 22H_2O + 2PO_4^{3-}$ | −23.1 [41] |
| $B(OH)_4^-$ | $B(OH)_3(aq) + H_2O = B(OH)_4^- + H^+$ | −9.24 [17] |
| $B_3O_3(OH)_4^-$ | $3B(OH)_3(aq) = B_3O_3(OH)_4^- + H^+ + 2H_2O$ | −7.528 |
| $B_4O_5(OH)_4^{2-}$ | $4B(OH)_3(aq) = B_4O_5(OH)_4^{2-} + 2H^+ + 3H_2O$ | −16.13 |
| $MgB(OH)_4^+$ | $Mg^{2+} + B(OH)_3 + H_2O = MgB(OH)_4^+ + H^+$ | −7.84 |
| $Mg(OH)2$ | $Mg(OH)_2 = Mg^{2+} + 2OH^-$ | −10.88 |

### 2.2. Hydration Algorithm

In this work, a state-oriented computer model is used to study the microstructure of MPC pastes performing on MATLAB [42]. In the model, there are five phases assumed in the hydration process: MgO particle, $KH_2PO_4$ (or $NH_4H_2PO_4$) particle, inner hydration product, outer hydration product, and large capillary pores. The input parameters of the model are the M/P ratio, W/B ratio, degree of hydration, molar density, and molar mass (as demonstrated in Table 3) of the reactants and hydration products.

**Table 3.** Molar density and molar mass of reactants and hydration products.

| Solid | Molar Mass (g/mol) | Molar Volume (cm³/mol) |
|---|---|---|
| $MgO$ | 40 | 11.3 |
| $KH_2PO_4$ | 136 | 58.2 |
| $NH_4H_2PO_4$ | 115 | 63.8 |
| $MgHPO_4 \cdot 3H_2O$ | 174 | 82.9 |
| $Mg_2KH(PO_4)_2 \cdot 15H_2O$ | 549 | 303.2 |
| $MgKPO_4 \cdot 6H_2O$ | 266 | 142.5 |
| $MgNH_4PO_4 \cdot 6H_2O$ | 245 | 143.5 |
| $(NH_4)_2Mg(HPO_4)_2 \cdot 4H_2O$ | 324 | 177.3 |

This model is based on the growing of spheres assumption; the reaction occurs on the surface of particles and the reactant is consumed layer by layer. The initial state of the hydration of MPC pastes is simulated by randomly distributing MgO and $KH_2PO_4$ (or $NH_4H_2PO_4$) spheres into a representative elementary volume (REV), which is a $100 \times 100 \times 100$ μm³ cubic with the periodic boundary condition. The particle size distributions of

MgO, $KH_2PO_4$ and $NH_4H_2PO_4$ are referred to in the experimental data [43,44]; therein, the particles with sizes smaller than 0.5 μm and larger than 45 μm are not considered for efficiency. The REV is discretized into $1000 \times 1000 \times 1000$ cubic voxels with a resolution of $0.1 \ \mu m^3$.

In the ultimate hydration state, $KH_2PO_4$ (or $NH_4H_2PO_4$) particles are fully reacted or dissolved in the solution, while the hydrated products constantly form around the MgO sphere layer by layer until the volume of hydrate is equal to the calculated one in this state. In this model, the layers whose thickness is equal to or smaller than the radius of MgO particles are called inner hydration products; otherwise, they are considered outer hydration products. In MKPC (or MAPC) paste, the inner and outer hydration products are the same, K-struvite (or struvite), so there is no distinction between these two products. The kinetics is not taken into consideration in this simulation.

The algorithm for calculating porosity is called the "Disk Filling Method", and it is based on the final hydration stage of the MKPC (or MAPC) paste. In this algorithm, a disc with an initial radius is used to fill the pores in the pastes. The radius of the disk continues to increase until the pores are completely filled. Thus, the corresponding pore volumes are considered to be equal to the volumes of these discs. Therefore, when the volume of REV is known, the cumulative porosity of the paste can be obtained.

### 2.3. Construction of Reaction System

In this work, the hydration simulation of MPC pastes is carried out at 25 °C and 0.1 MPa. The M/P ratio in MKPC pastes varies from 6 to 10, while W/B ratios are determined to be 0.2, 0.25, 0.3, and 0.5. The reaction step is set as 10,000, and the mix proportion of MKPC pastes is shown in Table 4. Moreover, the influence of boric acid as an admixture on the pH value of MKPC paste is analyzed. The mix proportion of reactants is 44.4 g of MgO, 55.6 g of $KH_2PO_4$, and 100 g of water. The amount of boric acid in the reactants is 0, 0.25, 1, and 2.5% of the mass of MgO and $KH_2PO_4$, as shown in Table 5. For the case of MAPC pastes, another 10 groups of reactants with different ratios are set to react, and the amount of MgO is fixed as 1 mol. The M/P ratio and W/B ratio of MAPC paste are shown in Table 6.

**Table 4.** Mix proportion of MKPC pastes.

| Paste | MgO/$KH_2PO_4$ Molar Ratio | W/B Mass Ratio |
|---|---|---|
| MP8H0.2K | 8 | 0.2 |
| MP8H0.3K | 8 | 0.25 |
| MP8H0.5K | 8 | 0.5 |
| MP6H0.25K | 6 | 0.25 |
| MP7H0.25K | 7 | 0.25 |
| MP8H0.25K | 8 | 0.25 |
| MP9H0.25K | 9 | 0.25 |
| MP10H0.25K | 10 | 0.25 |

**Table 5.** Mix proportion of MKPC pastes with boric acid.

| Paste | Boric Acid to Solid Mass Ratio (%) | MgO (g) | $KH_2PO_4$ (g) | Water (g) | Boric Acid (g) |
|---|---|---|---|---|---|
| MP2.7B0 | 0 | 44.4 | 55.6 | 100 | 0 |
| MP2.7B0.25 | 0.25 | 44.4 | 55.6 | 100 | 0.25 |
| MP2.7B1 | 1 | 44.4 | 55.6 | 100 | 1 |
| MP2.7B2.5 | 2.5 | 44.4 | 55.6 | 100 | 2.5 |

**Table 6.** Mix proportion of MAPC pastes.

| Paste | MgO/NH$_4$H$_2$PO$_4$ Molar Ratio | W/B Mass Ratio |
|---|---|---|
| MP3H0.5A | 3 | 0.5 |
| MP4H0.5A | 4 | 0.5 |
| MP4H1A | 4 | 1 |
| MP5H0.3A | 5 | 0.3 |
| MP5H0.5A | 5 | 0.5 |
| MP5H1A | 5 | 1 |
| MP8H0.3A | 8 | 0.3 |
| MP8H1A | 8 | 1 |

According to Table 2, the development of K-struvite in MKPC pastes and struvite in MAPC pastes must meet the following equalities:

$$logK \;<\; loga_{Mg^{2+}} + loga_{K^+} + loga_{PO_4^{3-}} + 6loga_{H_2O} \tag{3}$$

$$logK \;<\; loga_{Mg^{2+}} + loga_{NH_4^+} + loga_{PO_4^{3-}} + 6loga_{H_2O} \tag{4}$$

The hydration process of MPC pastes is simulated based on the minimal energy principle [45]. With initial settings of the reaction and input of the verified database, the ultimate equilibrium phase is obtained. In order to examine the accuracy of calculation for pH of the solution, MP8H5K is taken as an example; it is found that the calculated pH value agrees well with the experimental one from Xu et al. [15], as demonstrated in Figure 1.

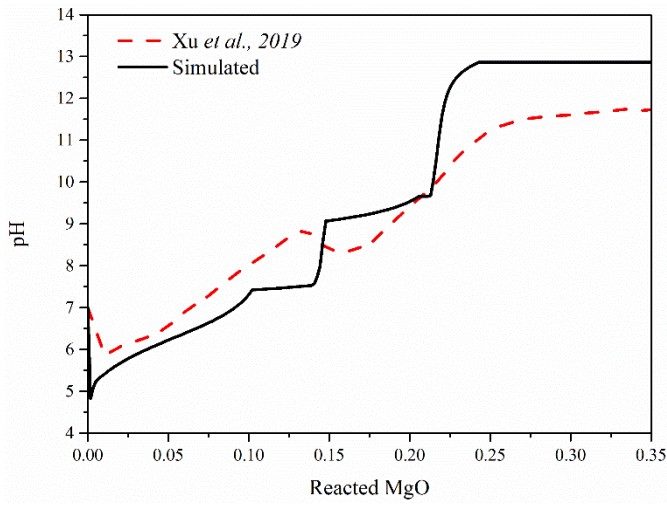

**Figure 1.** pH values of MP8H5 and experimental data in Ref. [15].

*2.4. Fractal Analysis of Pore Structure in MKPC Paste*

With respect to cement paste, fractal theory plays an important role in the pore structure analysis. Based on fractal theory, there is a relation among the number of pores whose diameters are larger than d, N(d), the probability density function of pore size distribution (f(d)), and the fractal dimension for pore size ($D_f$):

$$N(d) = (d_{max}/d)^{D_f} \tag{5}$$

$$f(d) = D_f \cdot d_{min}^{D_f} \cdot d^{-D_f-1} \tag{6}$$

where d, $d_{max}$, and $d_{min}$ are the given, maximum, and minimum pore size of the cement paste. Based on the image theory applied on a cross-section of a paste, $D_f$ is able to be obtained using box-counting method [46,47]. For each REV, ten cross-sections are analyzed based on the image analysis to calculate values of $D_f$, which are opposite numbers of slopes

of the paradigmatic logarithmic plot of cumulative pore account versus box size. The final $D_f$ of a paste is the average of ten values of $D_f$ of each cross-section.

## 3. Simulation Results and Analysis of MKPC Hydration

### 3.1. Influence of M/P and W/B Ratios on Hydration Products in MKPC Pastes

In this section, the simulation hydration results of MKPC pastes are presented. Figure 2 illustrates the evolution of the volume of hydration products and pH values with the reacted amount of MgO in MKPC pastes. From Figure 2, the hydration process of MKPC pastes can be identified as: firstly, $KH_2PO_4$ particles quickly dissolve in water and react with magnesium oxide to form $MgHPO_4 \cdot 3H_2O$. Then $Mg_2KH(PO_4)_2 \cdot 15H_2O$ appears and gradually dissolves in $MgHPO_4 \cdot 3H_2O$, and the main hydration product, K-struvite, is formed in large quantities. Additionally, the continuous reaction between MgO particle and water increases the pH value, which leads to the conversion from $MgHPO_4 \cdot 3H_2O$ to $Mg_2KH(PO_4)_2 \cdot 15H_2O$. Then the decomposition of $Mg_2KH(PO_4)_2 \cdot 15H_2O$ and the presence of K-struvite further increase the pH value. Xu et al. [15] considered the final hydration products of MKPC pastes were K-struvite and unreacted MgO with some intermediate products of $Mg_2KH(PO_4)_2 \cdot 15H_2O$, which shows a great agreement with the simulation results.

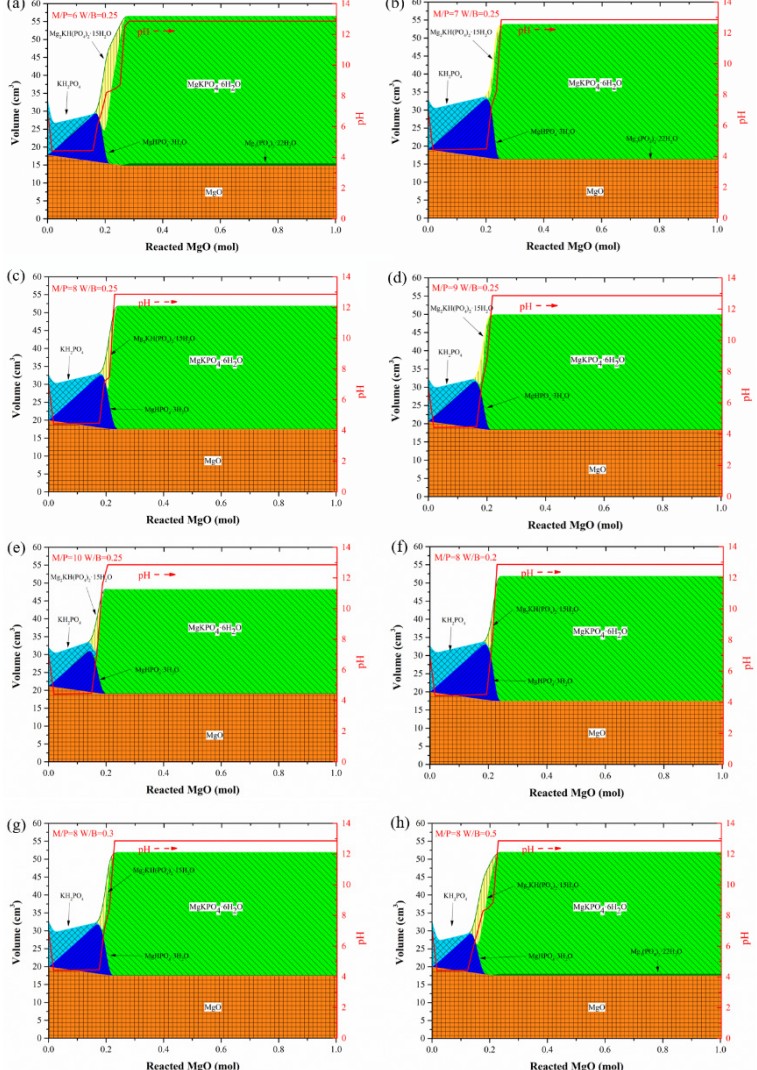

**Figure 2.** Evolution of volume of hydration products and pH values with the reacted amount of MgO in (**a**) MP6H0.25K; (**b**) MP7H0.25K; (**c**) MP8H0.25K; (**d**) MP9H0.25K; (**e**) MP10H0.25K; (**f**) MP8H0.2K; (**g**) MP8H0.3K; and (**h**) MP8H0.5K.

It was also found that the increase in the M/P ratio inhibits the formation of K-struvite of MKPC pastes as well as the formation of $Mg_2KH(PO_4)_2 \cdot 15H_2O$, while a low M/P ratio leads to the formation of $Mg_2(PO_4)_3 \cdot 22H_2O$. The increase in W/B ratio has little effect on K-struvite. The high W/B ratio also causes the formation of intermediate products such as $Mg_2(PO_4)_3 \cdot 22H_2O$ and $Mg_2KH(PO_4)_2 \cdot 15H_2O$ as well as inhibits the formation of $MgHPO_4 \cdot 3H_2O$. Moreover, no significant differences in the ultimate pH values are found among these eight pastes in Figure 2. Moreover, an increase of pH value in the solution implies that main hydration products are formed in large quantities, and intermediate products simultaneously disappear. The same phenomenon appears for the case of Rouzic et al. and Lahalle et al. [16,48].

### 3.2. Influence of Boric Acid on Hydration Products and pH in MKPC Pastes

The evolution of the amount of hydration products with the reacted amount of MgO in each mix proportion of Table 5 is shown in Figure 3, while the corresponding pH values of the pastes are compared in Figure 4. Expectedly, the addition of boric acid significantly decreases the pH value of the paste, inhibits the appearance of K-struvite, and promotes the formation of $Mg_2(PO_4)_3 \cdot 22H_2O$. A similar phenomenon was observed by Lahalle et al.: boric acid does not slow down the initial dissolution of the reactants (MgO and $KH_2PO_4$), but it delays the precipitation of hydration products. In order to illustrate the influence of boric acid on the hydration of MPCs, the evolution of the molarity of ions and pH value in MP2.7B0.25 solution and comparison of the molarity of MP2.7B0.25 and MP2.7B0 are shown in Figures 5 and 6, respectively. It can be found that there are five stages in the evolution of the molarity of the ions in the MKPC solution with boric acid, which are shown in Table 7.

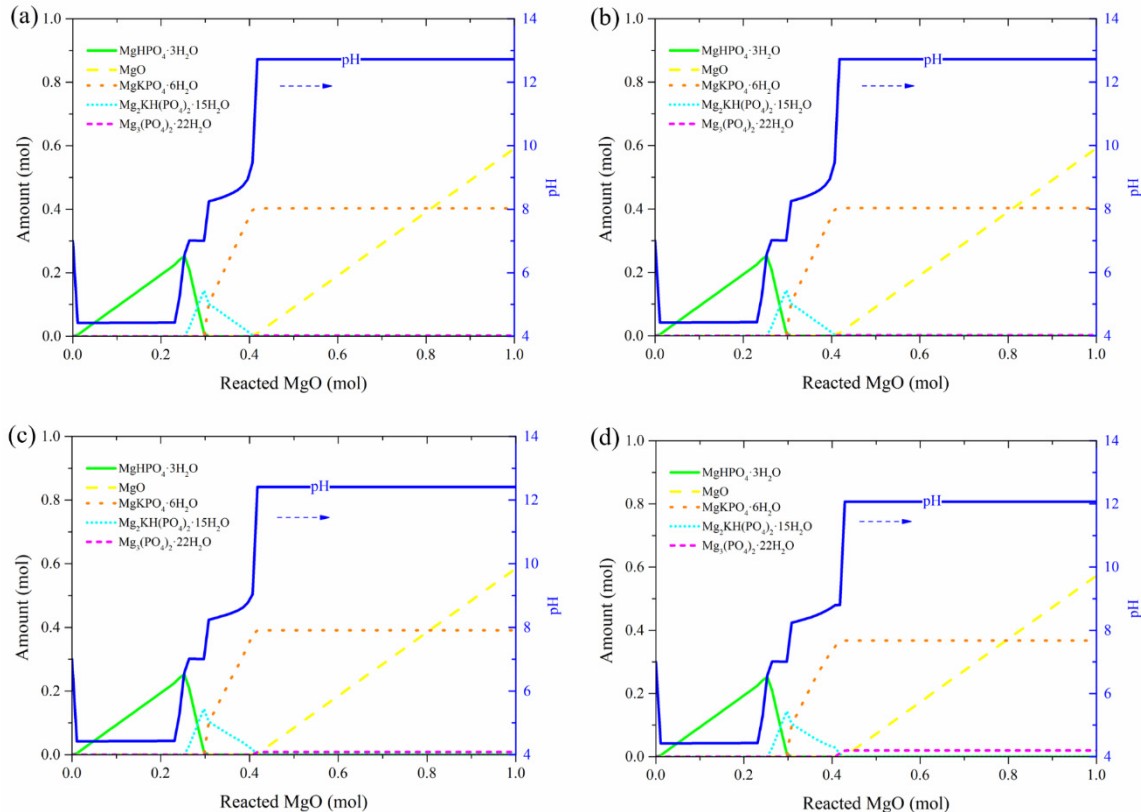

**Figure 3.** Evolution of amount of hydration products with reacted amount of MgO in (**a**) MP2.7B0K; (**b**) MP2.7B0.25K; (**c**) MP2.7B1K; and (**d**) MP2.7B2.5K.

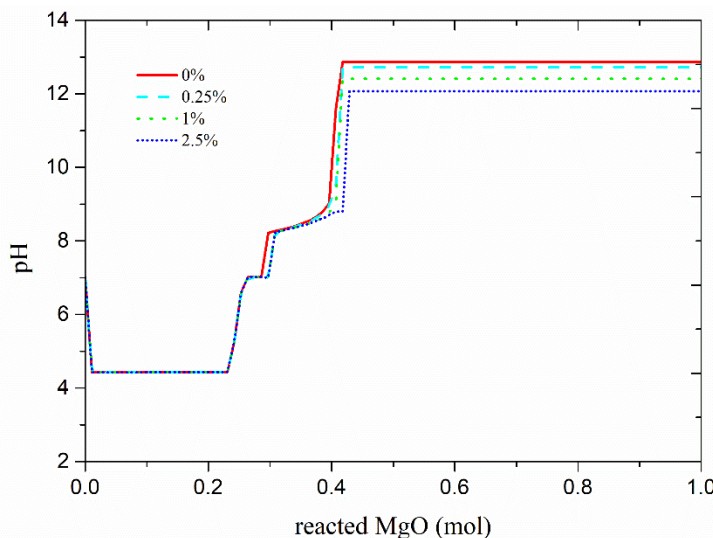

**Figure 4.** Comparison of pH in MKPC pastes with different amounts of boric acid.

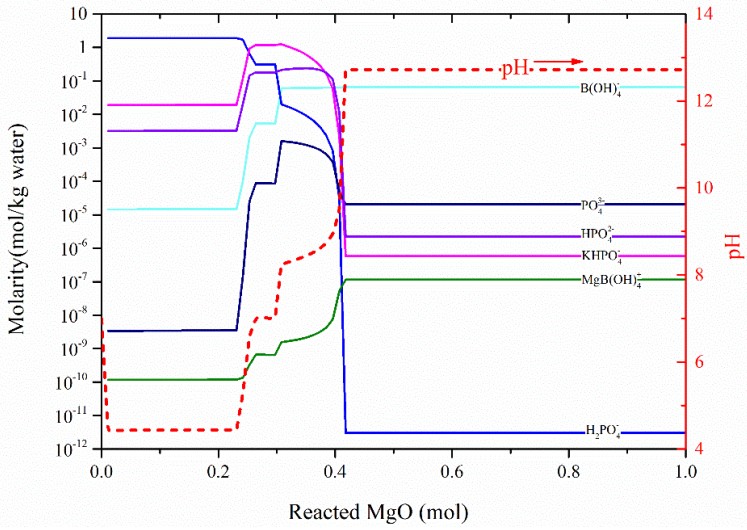

**Figure 5.** Evolution of the molarity of ions and pH value in MP2.7B0.25 solution with reacted amount of MgO.

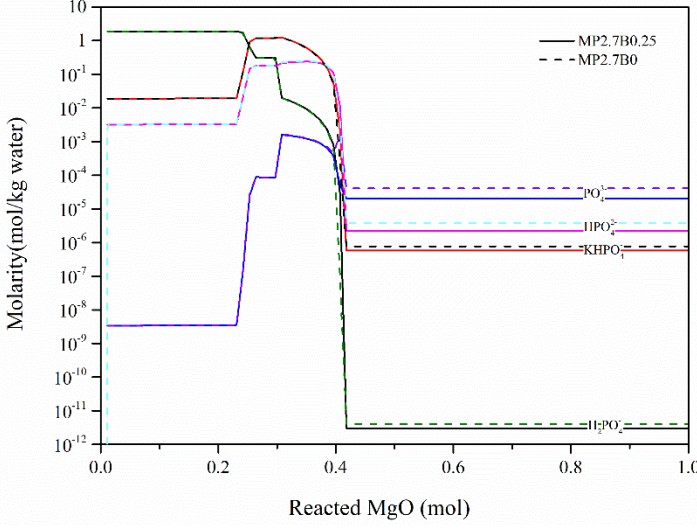

**Figure 6.** Comparison of the molarity of MP2.7B0.25 and MP2.7B0 during the hydration.

**Table 7.** Stages in the evolution of the molarity of the ions in the MKPC solution with boric acid.

| Stage | Reacted MgO (mol) | pH | Molarity | | | | | |
|-------|-------------------|-----|----------|---------|---------|----------|-----------|---------|
| | | | $B(OH)_4^-$ | $PO_4^{3-}$ | $HPO_4^{2-}$ | $KHPO_4^-$ | $MgB(OH)_4^+$ | $H_2PO_4^-$ |
| Stage I | 0–0.231 | 4.4 | - | - | - | - | - | - |
| Stage II | 0.231–0.264 | 4.4–7 | increased | increased | increased | increased | increased | decreased |
| Stage III | 0.264–0.297 | 7 | - | - | - | - | - | - |
| Stage IV | 0.297–0.363 | 7–8.5 | increased | increased | - | - | increased | decreased |
| Stage V | 0.363–0.418 | 8.5–12.7 | - | decreased | decreased | decreased | increased | decreased |

From Table 7 and Figure 3, the hydration process of the MKPC with boric acid can be figured out to some extent. Firstly, MgO particles react with $KH_2PO_4$ particles and form $MgHPO_4 \cdot 3H_2O$, and the boric acid dissolves in the solution, forming $B(OH)_4^-$ and a small amount of $MgB(OH)_4^+$. Then, the excessive MgO raises the pH of the solution from 4.4 to 7 with part of $H_2PO_4^-$ changing into $PO_4^{3-}$, $HPO_4^{2-}$, and $KHPO_4^-$. At the same time, more and more boric acid dissolves into $B(OH)_4^-$ and $MgB(OH)_4^+$. When pH reaches seven, there is a plateau because $MgHPO_4 \cdot 3H_2O$ gradually changes into $Mg_2KH(PO_4)_2 \cdot 15H_2O$. When the amount of $Mg_2KH(PO_4)_2 \cdot 15H_2O$ gets to a peak, the pH continuously raises, and K-struvite begins to appear. Because of the formation of $MgB(OH)_4^+$, the amount of K-struvite rises slowly, and the final pH value of the MKPC solution is lower with boric acid than without boric acid, while the ultimate amount of K-struvite is not significantly influenced. It can be also found that the boric acid does not slow down the initial dissolution of MgO and KH2PO4, but $MgB(OH)_4^+$ together with $B(OH)_4^-$ slows down the formation of K-struvite. The increase of $B(OH)_4^-$ affects the molarity of phosphate, while lots of magnesium is combined to form $MgB(OH)_4^+$, which favors the formation of $Mg_2(PO_4)_3 \cdot 22H_2O$ and retards the formation of K-struvite.

*3.3. Pore structure of MKPC Pastes*

The simulated microstructure of different MKPC pastes in the initial and ultimate hydration state is shown in Figure 7. Different phases are represented by different colors: orange, light blue, green, and dark blue represent the MgO particle, $KH_2PO_4$ particle, K-struvite, and pore (or water). From Figure 7, it is anticipated that $KH_2PO_4$ particles are totally dissolved in the water or completely reacted in the paste, and MgO particles are redistributed throughout the REV due to the long-term reaction. It can be also found that small MgO particles are completely hydrated due to the fast reaction rate, while the hydration part of large MgO particles is controlled by the water diffusion in the deposited hydrates covering the surface of MgO particles. Therefore, once the external hydrate is decomposed into water and MgO particles are exposed to water, the internal non-hydrated MgO part may be hydrated again. In addition, during the hydration process, with the increase of the ion concentration in the solution, K-struvite is also gradually formed around the particles, and together with MgO and the hydration products on its surface, it forms the skeleton of the cement paste, contributing to the increase of paste strength. The original pores of MKPC pastes in the initial hydration state are gradually filled by hydration products, and the large capillary pores in the ultimate hydration state are the voids left by the cement particles that are not filled by hydration products and the ones by the water-filled spaces.

It is generally believed that cement-based materials have a porous structure, and their porosities significantly affect the mechanical properties and durability of the materials [21]. Therefore, porosity, as the most commonly used indicator, is adopted in this work to evaluate MKPC pastes. The porosity of MP8H0.25K is compared with experimental data determined by the mercury intrusion porosimetry test to verify the reliability of the porosity derivation, which is presented in Figure 8.

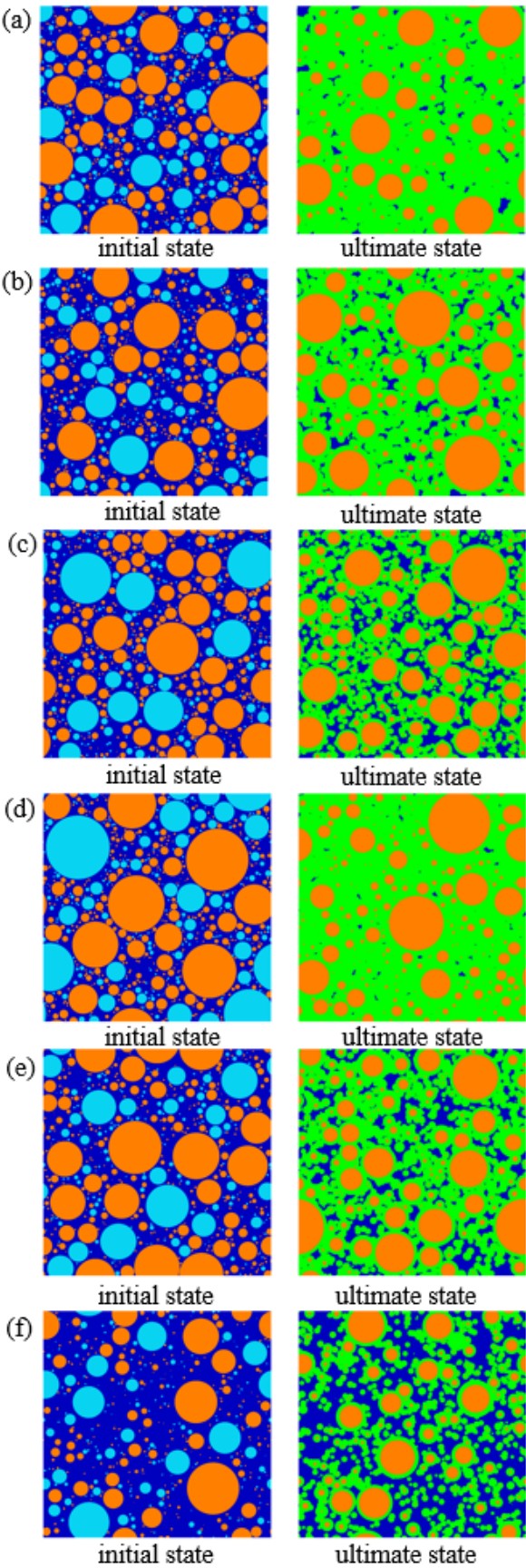

**Figure 7.** Simulated microstructure of (**a**) MP6H0.25K; (**b**) MP8H0.25K; (**c**) MP10H0.25K; (**d**) MP8H0.2K; (**e**) MP8H0.3K; and (**f**) MP8H0.5K.

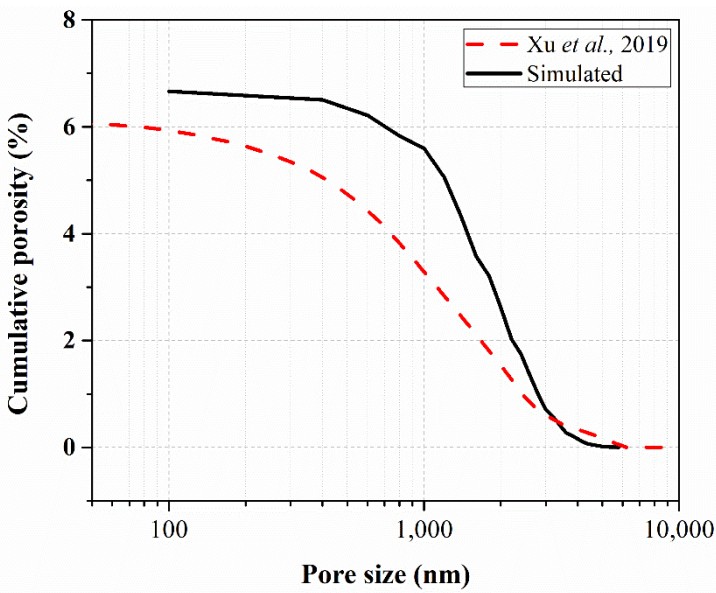

**Figure 8.** Porosity of simulated and experimental data in Ref. [15] for MP8H0.25K.

Additionally, the comparison of cumulative porosity in different MKPC pastes is shown in Figure 9. It is apparent from the figures that the cumulative porosity of MKPC pastes decreases from 18.3% to 3.77% with the decrease of M/P ratio from 10 to 6 when W/B ratio is fixed at 0.25. This result is potentially attributed to the fact that the increase of $KH_2PO_4$ particle in the paste leads to a large amount of K-struvite and thus a small porosity. On the contrary, the increase of the W/B ratio casts a negative impact on the porosity evolution of MKPC pastes. In general, the pore size of the MKPC paste is between 100 and 5000 nm, which is in agreement with Ma et al., and with the increase of W/B and M/P ratios, the proportion of large pores remarkably increases.

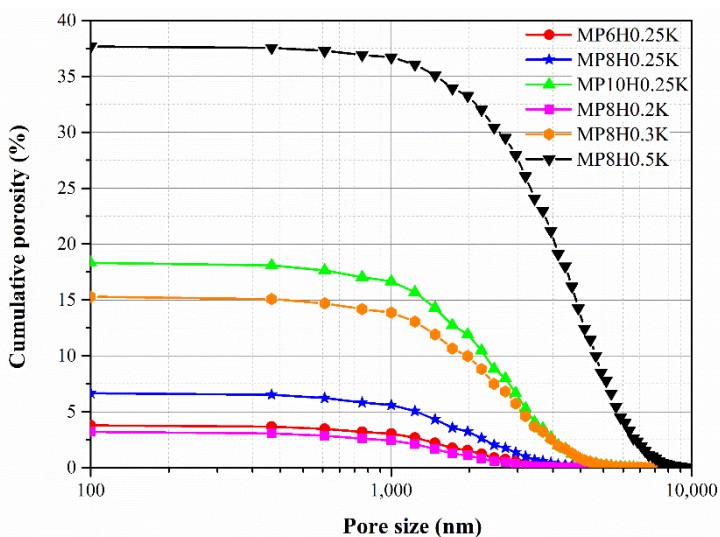

**Figure 9.** Comparison of cumulative porosity in different MKPC pastes.

### 3.4. Analysis of Fractal Features of MKPC Pastes

Based on pore structure analysis and Equations (3) and (4), an analysis of fractal features of MKPC pastes was performed. The calculated $D_f$ in MKPC pastes are shown in Figure 10, which is in good agreement with Yu et al. and Tang et al. [49], while the difference between the simulation data and experimental data is potentially caused by the difference in pore size distribution. It is presented in Figure 10 that the $D_f$ in the MKPC pastes varies between 1.4 and 2, which is consistent with the literature. Additionally, the $D_f$

of MKPC paste increases as the porosity of the pastes increases, and the larger the porosity grows, the slower $D_f$ increases. In this work, the $D_f$ of MKPC pastes in the ultimate stage is smaller than that in initial stage because the hydration products of MKPC pastes fill in the pores in the initial stage, as the small $D_f$ attributes to the small porosity in the ultimate stage.

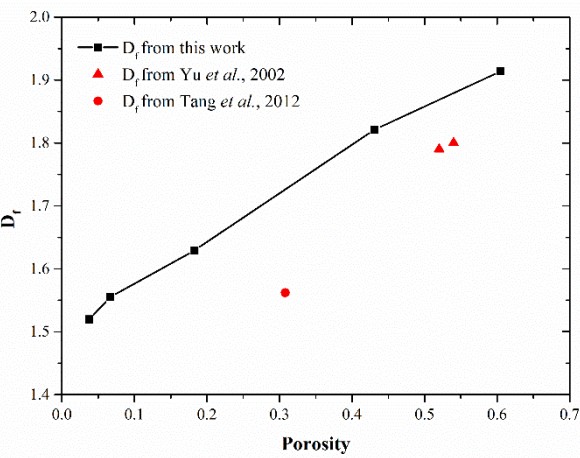

**Figure 10.** Evolution of $D_f$ with porosity from this work and Refs. [49,50].

Moreover, the $D_f$ of MKPC pastes in this work is related to the M/P ratio and W/B ratio. As shown in Figures 9 and 10, a small M/P ratio as well as a small W/B ratio gives birth to a small $D_f$. This phenomenon may be explained, as the small M/P ratio leads to a large amount of K-struvite and small capillary pores, which results in a small $D_f$, while the small W/B ratio enables a small fraction of water in the total volume of MKPC pastes in the initial stage, resulting in a relatively large faction of small pores rather than large pores.

## 4. Simulation Results and Analysis of MAPC Hydration

### 4.1. Influence of M/P and W/B Ratios on Hydration Products in MAPC Pastes

Figure 11 illustrates the evolution of volume of hydration product and pH value with the reacted amount of MgO in the MAPC pastes. In Figure 11, yellow, green, blue, and brown colors represent the struvite, $(NH_4)_2Mg(HPO_4)_2 \cdot 4H_2O$, MgO, and $NH_4H_2PO_4$. The initial pH value of MAPC pastes is 4.59, and with the addition of MgO, the pH expectedly increases. Compared with Figure 11, the influence of M/P ratio has a more profound effect on the change of pH value than that of W/B ratio in the paste. Due to the presence of MgO, the pH in the system has a short plateau, and then rapidly rises to around 6 and 12 for second and third plateaus. Compared with Figure 11c,e,g, when M/P ratio is fixed, W/B ratio hardly affects the type and volume of hydration products.

$MgHPO_4 \cdot 3H_2O$ is basically not generated in Figure 11 above, which is in line with the report in the literature [40]. $(NH_4)_2Mg(HPO_4)_2 \cdot 4H_2O$ and struvite appear in sequence when the reaction progresses in a high M/P ratio. However, when M/P ratio is equal to 0.5, the final product is only $(NH_4)_2Mg(HPO_4)_2 \cdot 4H_2O$. Accordingly, when M/P ratio is low, the intermediate product $(NH_4)_2Mg(HPO_4)_2 \cdot 4H_2O$ is generated in a large amount, while a high M/P ratio in turn inhibits the formation of intermediate products and promotes the formation of struvite. In addition, in Figure 11, the pH value of the paste remains stable when $(NH_4)_2Mg(HPO_4)_2 \cdot 4H_2O$ turns into struvite. The reacted MgO constantly releases $Mg^+$ into the system and directly reacts with $(NH_4)_2Mg(HPO_4)_2 \cdot 4H_2O$, forming struvite without releasing ammonium ion and phosphate into the system. While the amount of struvite in the system reaches a peak, the excessive MgO raises the pH of the solution rapidly.

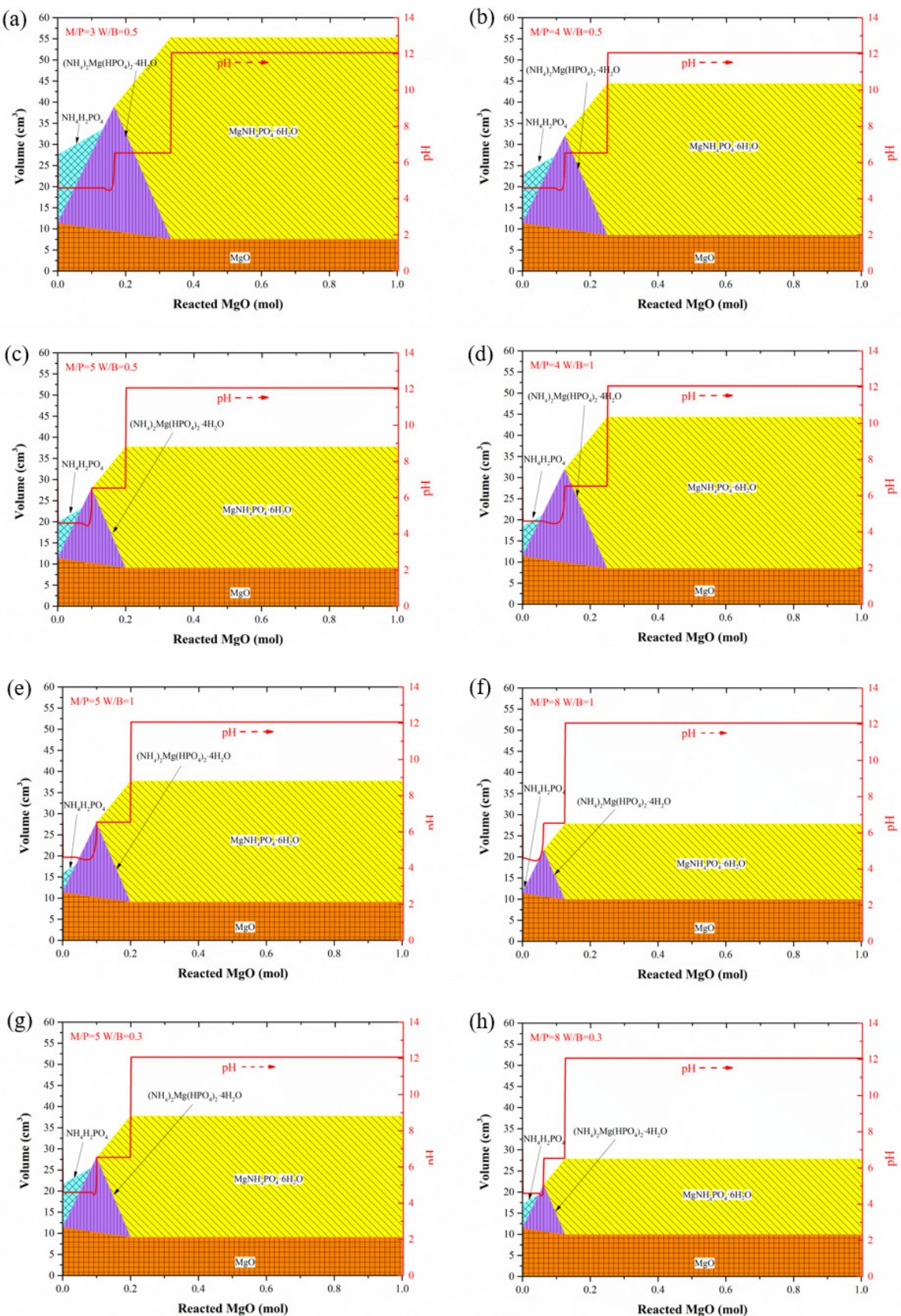

**Figure 11.** Evolution of volume of hydration products and pH values with reacted amount of MgO in (**a**) MP3H0.5A; (**b**) MP4H0.5A; (**c**) MP5H0.5A; (**d**) MP4H1A; (**e**) MP5H1A; (**f**) MP8H1A; (**g**) MP5H0.3A; and (**h**) MP8H0.3A.

### 4.2. Pore Structure of MAPC Pastes

In Figure 12, the simulated microstructure of different MAPC pastes in the initial and ultimate hydration state is presented. Yellow, light blue, and green spheres represent MgO, $NH_4H_2PO_4$, and struvite, while dark blue is for water and pores. As shown in Figure 12, in the ultimate state, the reaction occurs on the surface of MgO that is often covered by struvite and hardly can be completely hydrated, and the excessive MgO particles may act as aggregates, enhancing the stability and strength of the paste structure. It is found that when the M/P ratio is fixed, within a certain range, the smaller the W/B ratio is, the denser the formed paste structure is, which is in line with Wang et al. [51]. These trends are similar with ones of MKPC pastes.

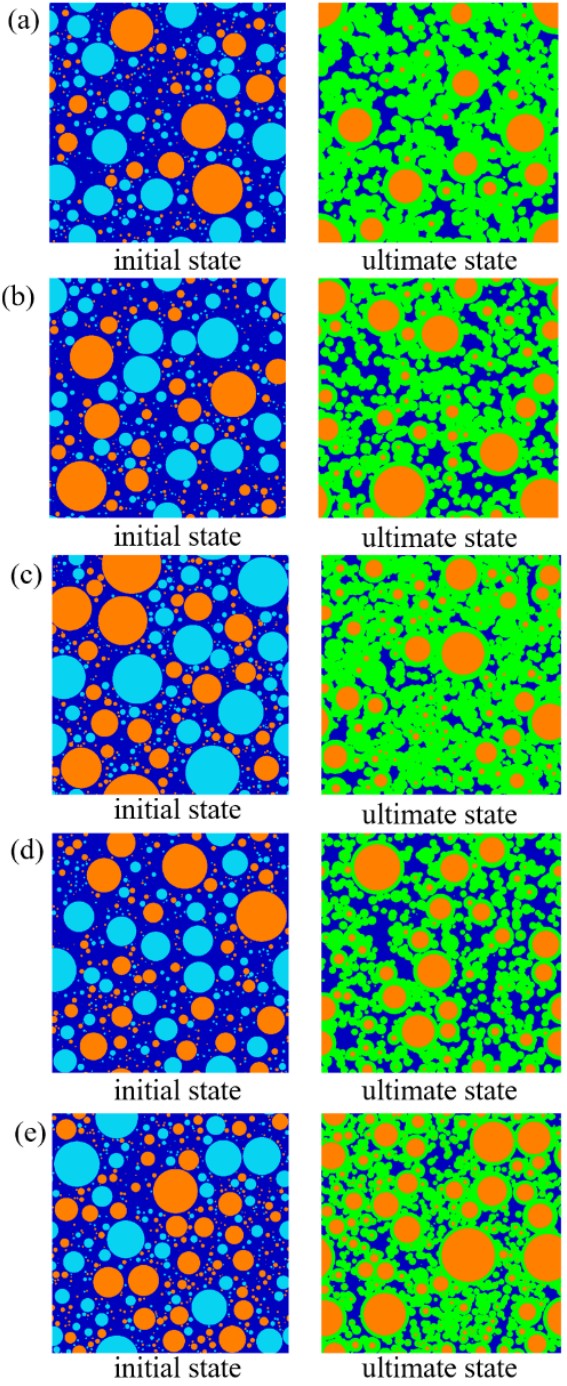

**Figure 12.** Simulated microstructure of (**a**) MP3H0.5A; (**b**) MP4H0.5A; (**c**) MP5H0.3A; (**d**) MP5H0.5A; and (**e**) MP8H0.3A.

Figure 13 is the comparison of cumulative porosity in different MAPC pastes. When W/B ratio is 0.3, the cumulative porosities of MP5H0.3A and MP8H0.3A are smaller than 20%. Relatively speaking, the sequence of cumulative porosity is MP4H0.5A > MP5H0.5A > MP3H0.5A > MP8H0.3A > MP5H0.3A. In practical application, when the M/P ratio is larger than one, it is conductive to the formation of hydration products. It can be found from Figure 13 that when M/P ratio is fixed at five, the hydration product structure of MP5H0.3A is the densest, a small W/B ratio obviously leads to a compact structure of the final cement paste, and vice versa. When W/B ratio is fixed, as M/P ratio increases from three to five, the cumulative porosity increases from 20.79% to 29.75%, which may be attributed by the decrease of the volume fraction of struvite in MAPC pastes. According to Figure 12, as the ultimate state of MAPC pastes is composed of struvite and excessive MgO, the decrease of M/P ratio significantly increases the volume of the whole paste as well as enlarges the volume fraction of struvite inside. Moreover, the fraction of small pores in the cumulative porosity of pastes with a low W/B ratio is larger than that of pastes with a high W/B ratio; MgO particles are remarkably separated for the case of a high W/B ratio and beneficial for the formation of pores with a large size within struvite.

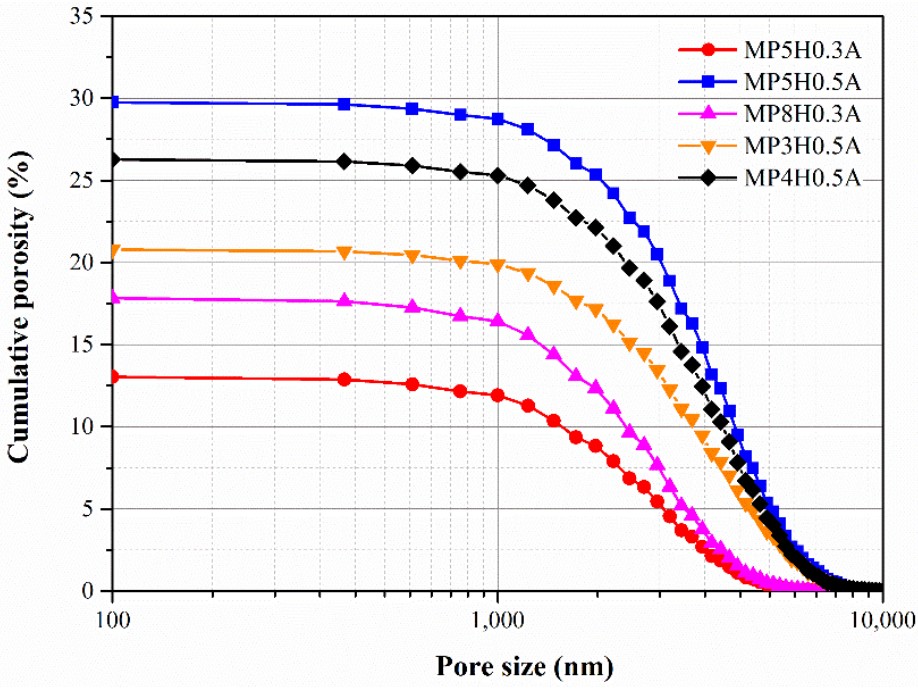

**Figure 13.** Comparison of cumulative porosity in different MAPC pastes.

### 4.3. Comparison between MKPC System and MAPC System

Due to the presence of ammonium ion, the pH of MAPC paste is lower than that of MKPC one, and it can be found that the hydration process of MKPC paste is more complicated than that of the MAPC one. More intermediate hydration products show up in the hydration process of MKPC pastes than those of MAPC ones. For example, $MgHPO_4 \cdot 3H_2O$ hardly appears in the hydration process of MAPC pastes, while it is the main intermediate product in the MKPC one. A similar phenomenon was found by Han et al. and Zhang et al. [41]. Generally speaking, with respect to the same condition and mix proportion, the cumulative porosity of MKPC pastes is lower than MAPC ones. The comparison of porosities among simulated and experimental results is shown in Table 8. Although the molar volume of struvite and K-struvite are close, the volume fraction of K-struvite in MKPC paste is larger than that of struvite in MAPC ones, which leads to a smaller porosity in MKPC paste at the same mix proportion.

**Table 8.** Comparison of porosities among simulated and experimental results.

| Paste | Porosity by Simulation (%) | Porosity by Experiment (%) | Difference (%) |
|---|---|---|---|
| MP6H0.25K | 3.77 | - | - |
| MP6H0.2K | - | 5.46 [48] | - |
| MP8H0.25K | 6.66 | 6.06 [15] | 9.9 |
| MP8H0.5K | 37.67 | 40.24 [21] | 6.4 |
| MP10H0.25K | 18.31 | - | - |
| MP10H0.5K | - | 42.69 [21] | - |
| MP4H0.5A | 26.15 | - | - |
| MP5H0.3A | 12.88 | - | - |
| MP5H0.5A | 29.76 | - | - |
| MP4H0.22A | - | 6.18 [52] | - |

## 5. Conclusions and Prospects

Several conclusions can be drawn:

(1) M/P and W/B ratios significantly affect the hydration products in MKPC pastes. In the hydration process of MKPC pastes, intermediate products, such as $MgHPO_4 \cdot 3H_2O$ and $Mg_2KH(PO_4)_2 \cdot 15H_2O$, form first before the appearance of the main hydration product, K-struvite, and $Mg_2(PO4)_3 \cdot 22H_2O$ is generated in a small amount together with K-struvite. Additionally, the increase in the M/P ratio inhibits the formation of K-struvite, and a low M/P ratio facilitates the formation of intermediate products. A high W/B ratio also causes the formation of intermediate products and accelerates the formation of K-struvite. As for MAPC pastes, M/P and W/B ratios play the same role in the influence of the main hydration product. With the hydration process, the intermediate product $(NH_4)_2Mg(HPO_4)_2 \cdot 4H_2O$ is gradually replaced by struvite.

(2) The addition of boric acid significantly reduces the pH of the reaction, inhibits the formation of K-struvite, as well as promotes the formation of intermediate products. The boric acid does not slow down the initial dissolution of MgO and KH2PO4, but $MgB(OH)_4^+$ together with $B(OH)_4^-$ slows down the formation of K-struvite and lowers the pH value of the solution.

(3) From the initial and final hydration states of MKPC and MAPC pastes, it can be found that in the hydration process, the small MgO particles are entirely hydrated due to the quick reaction, while the large ones are covered by inner hydration products and outer hydration products. According to the porosity analysis conducted based on hydration state, the increase in M/P ratio in the paste leads to a large volume fraction of K-struvite/struvite, which is conductive to a small porosity of the paste. In contrast, a high W/B ratio increases the porosity of MPC pastes. With the increase of M/P and W/B ratios, there is a significant growth in the fraction of large pores. Additionally, fractal analysis in this work reveals that the $D_f$ of MKPC pastes is positively proportional to the porosity and small M/P ratios, and small W/B ratios are beneficial for reducing the $D_f$ of MKPC pastes.

**Author Contributions:** Conceptualization, S.T. and J.H.; Data curation, Y.P. and C.T.; Formal analysis, Y.P., C.T. and Y.L.; Funding acquisition, S.T.; Methodology, Y.P., S.T., L.W. and J.H.; Project administration, S.T.; Software, Y.P., J.H. and C.T.; Validation, Y.P.; Visualization, Y.P.; Writing—original draft, Y.P.; Writing—review and editing, Y.P. and S.T. All authors have read and agreed to the published version of the manuscript.

**Funding:** This work was supported by the Opening Funds of the State Key Laboratory of Building Safety and Built Environment and National Engineering Research Center of Building Technology (Grant No. BSBE2020-1) and the Yangtze River Water Science Research Joint Fund Key Project of the National Natural Science Foundation of China (Grant No. U2040222).

**Institutional Review Board Statement:** Studies did not involve humans.

**Informed Consent Statement:** Studies did not involve humans.

**Data Availability Statement:** The data that support the findings of this study are available from the corresponding author upon reasonable request.

**Acknowledgments:** The authors would like to thank all the anonymous referees for their constructive comments and suggestions.

**Conflicts of Interest:** The authors declare no conflict of interest.

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
