# Peer review of "Fractal Analysis on Pore Structure and Modeling of Hydration of Magnesium Phosphate Cement Paste"

_fractalfract, doi:10.3390/fractalfract6060337_

Round 1
Reviewer 1 Report
The article entitled "Fractal analysis on pore structure and modeling of hydration of magnesium phosphate cement paste" deals with MPC paste and evaluate the thermodynamic data in specific conditions and an reaction model established.
In Methods section all the necessary parameters and methodology provided and in the results In results section the results presented in detailed and compared with the existing literature (The comparison could be appears in a Discussion section)
I propose the acceptance of the article as it is.
Author Response
The main corrections in this manuscript are as follows:
- In Line 37, reference [7] and [8] are added.
- In Line 69, reference [24] is added.
- In Lines 93-95, detail reason for using MINTEQ database is added.
- In Line 106. reference for MATLAB is added.
- In Lines 130-135, a new paragraph is added to illustrated how the porosity of the paste is obtained in the fractal analysis.
- In Lines 142-143, “KH2PO4” is corrected to “KH2PO4”.
- In Table 5, the MKPC pastes MP2.7B0 is now placed at the top of the table.
- In Figs 1, 3, 4, 8, 9, and 13, lines are replotted for easy read in black and white.
- In Figs 2 and 11, colors in both figures are used to ensure the consistence.
- In Lines 254-256, the illustration for each color in Figures 7 is added.
- In Lines 256-275, the paragraph is rewritten to illustrate the concept of five phases especially large capillary pores along with how they form in the hydration process.
- In Line 288, “MP8H0.25” is corrected to “MP8H0.25K”.
- Figs 7 and 12 are replotted.
- In Line 304, new reference [51] is added and Fig 10 is replotted with new reference data.
- In Lines 304-306, the explanation for the difference of data is added.
- In Line 353, the explanation for colors is corrected.
- Table 8 is refined to show clear difference between porosity by simulation and experiment.
Point 1: The article entitled "Fractal analysis on pore structure and modeling of hydration of magnesium phosphate cement paste" deals with MPC paste and evaluate the thermodynamic data in specific conditions and a reaction model established.
In Methods section all the necessary parameters and methodology provided and in the results. In results section the results presented in detailed and compared with the existing literature (The comparison could be appearing in a Discussion section)
I propose the acceptance of the article as it is.
Response 1: Thanks for reviewer’s positive comments.
Reviewer 2 Report
The manuscript entitled “Fractal analysis on pore structure and modeling of hydration of magnesium phosphate cement paste” requires minor revision before publication. The author mainly described a thermodynamic simulation on the hydration of magnesium phosphate cement paste. This work applies fractal analysis for the pore microstructure of the MPC pastes with variant M/P or W/B ratios. PHREEQC is the software deployed on the state-oriented computer model. Although the manuscript is ready to be accepted, several suggestions were made to authors for revision. Please find the comments below.
(1) The MINTEQA2 V.4 database was used in the study for the formation of K-struvite or struvite phase. However, Blanc cement chemistry database was noted in the Table 1. It is mainly collected by detailed C-S-H data of different forms. How about the simulation results from Blanc cement chemistry database?
(2) What is a state-oriented computer model applied in the study? Please add an illustration for the concept of five phases especially the large capillary pores.
(3) At line 131 and 132, please note the presentation of KH2PO4.
(4) In table 5, mix proportion of MKPC pastes (MP2.7) should be put into an order with boric acid. The native MKPC pastes MP2.7B0 was named at the bottom of the table now.
(5) Where does the pore come from in a fractal analysis based on thermodynamic simulation? It is well to show the equation 5 and 6 in the manuscript. But, there is a gap or an appropriate reference need to be filled. A figure may be suggested to show the pore structure according to the fractal theory.
(6) Why the color usage is different in Figure 2 and Figure 11? At lease, the color should be the same for MgO.
(7) What is the representative meaning of each color in Figure 7 and Figure 12? Should it be a correspondence of Figure 2 and Figure 11 and an extra color for pore sites?
Reviewer 3 Report
The article titled "Fractal analysis on pore structure and modeling of hydration of magnesium phosphate cement paste" by Yuxiang Peng et al studies the influence of the magnesium-phosphorus molar (M/P) ratio and water-to-binder (W/B) ratio on the hydration product of Potassium magnesium phosphate cement (MKPC) paste, explored in this study by the thermodynamic simulation. The mechanical properties of the paste are determined by its complex pore structure and is therefore consider herein as a porous composite material. Some studies quoted by the authors determined, from fractal analysis of the paste, that the fractal dimension was a vital parameter related to pore volume. In this study, fractal analysis was applied to the simulated pore structures of the MPC pastes to evaluate the relationship between their microstructure and M/P or W/B ratios.
General Remarque
While the fractal dimension is obtained using the box-count method on the simulated microstructures of the MKPC pastes (Fig. 7), how porosity is obtained is not explained. The comparisons of the obtained porosity against the ones from reference 13 have only in common, the shape, otherwise the difference between the two is very significant, therefore requires an explanation. As for the fractal dimension results, the same remark can be applied and worse, there are only two points on the curve for comparison (Fig. 10). This is unacceptable. More data points and better comparison are required for the results to be acceptable.
Detailed remarks
1. Please give the references of the simulation software, the name of the developer/company and addresses.
2. Same remark for the databases.
3. Page 7, line 162, please provide detailed information for the box-counting software.
4. For different curves, please plot using different line styles for ease of reading when print is in black and white (Fig3, 4, 5, 8, ...).
5. Page 20, Table 8. The table of porosties should be redone, e.g.,
Paste Simulation Experiment & Ref difference %
MP6H0.2K 5.46 7.59 [18] 28
Round 2
Reviewer 3 Report
The authors have satisfactorily addressed the concerns expressed and made the necessary changes to the manuscript.